# NOX2-Induced High Glycolytic Activity Contributes to the Gain of COL5A1-Mediated Mesenchymal Phenotype in GBM

**DOI:** 10.3390/cancers14030516

**Published:** 2022-01-20

**Authors:** Youngjoon Park, Minwoo Park, Junhyung Kim, Juwon Ahn, Jeongmin Sim, Ji-In Bang, Jinhyung Heo, Hyejeong Choi, Kyunggi Cho, Mihye Lee, Jong-Seok Moon, Jaejoon Lim

**Affiliations:** 1Bundang CHA Medical Center, Department of Neurosurgery, CHA University, Yatap-dong 59, Seongnam 13496, Korea; yjparkep@chauniv.ac.kr (Y.P.); eugene@chauniv.ac.kr (J.A.); simti123@naver.com (J.S.); sandori50@chamc.co.kr (K.C.); 2Department of Integrated Biomedical Science, Soonchunhyang Institute of Medi-bio Science (SIMS), Soonchunhyang University, Cheonan 31151, Korea; pmw0269@sch.ac.kr (M.P.); hotdog58@naver.com (J.K.); mihyelee@sch.ac.kr (M.L.); 3Bundang CHA Medical Center, Department of Nuclear Medicine, CHA University, Yatap-dong 59, Seongnam 13496, Korea; bang14@cha.ac.kr; 4Bundang CHA Medical Center, Department of Pathology, CHA University, Yatap-dong 59, Seongnam 13496, Korea; sacrum77@chamc.co.kr; 5Bundang CHA Medical Center, Department of Radiology, CHA University, Yatap-dong 59, Seongnam 13496, Korea; isahj@cha.ac.kr

**Keywords:** NOX2, glycolysis, COL5A1, mesenchymal phenotype, GBM

## Abstract

**Simple Summary:**

Glioblastoma multiforme (GBM) is the most aggressive type of glioma and exhibits extensive heterogeneity and poor prognosis with a high recurrence rate. Among the genetic alterations in GBM with different phenotypic states, a mesenchymal subtype has been associated with a worse outcome in patients with GBM. The mechanisms for the gain of the mesenchymal subtype in GBM remain unclear. Our aim was to investigate whether NOX2-induced high glycolytic activity could contribute to the gain of the mesenchymal phenotype in GBM. We revealed that NOX2-induced high glycolytic activity can induce the gain of the COL5A1-mediated mesenchymal phenotype in GBM. Our findings will provide the molecular mechanism by which NOX2 contributes to the gain of mesenchymal phenotype in GBM.

**Abstract:**

The alteration of the cellular metabolism is a hallmark of glioma. The high glycolytic phenotype is a critical factor in the pathogenesis of high-grade glioma, including glioblastoma multiforme (GBM). GBM has been stratified into three subtypes as the proneural, mesenchymal, and classical subtypes. High glycolytic activity was found in mesenchymal GBM relative to proneural GBM. NADPH oxidase 2 (NOX2) has been linked to cellular metabolism and epithelial-mesenchymal transition (EMT) in tumors. The role of NOX2 in the regulation of the high glycolytic phenotype and the gain of the mesenchymal subtype in glioma remain unclear. Here, our results show that the levels of NOX2 were elevated in patients with GBM. NOX2 induces hexokinase 2 (HK2)-dependent high glycolytic activity in U87MG glioma cells. High levels of NOX2 are correlated with high levels of HK2 and glucose uptake in patients with GBM relative to benign glioma. Moreover, NOX2 increases the expression of mesenchymal-subtype-related genes, including *COL5A1* and *FN1* in U87MG glioma cells. High levels of NOX2 are correlated with high levels of COL5A1 and the accumulation of extracellular matrix (ECM) in patients with GBM relative to benign glioma. Furthermore, high levels of HK2 are correlated with high levels of COL5A1 in patients with GBM relative to benign glioma. Our results suggest that NOX2-induced high glycolytic activity contributes to the gain of the COL5A1-mediated mesenchymal phenotype in GBM.

## 1. Introduction

Glioma is a malignant brain tumor with the highest incidence among central nervous system (CNS) tumors [1,2]. According to the 2016 WHO brain tumor classification criteria, glioma is divided into biologically benign (grade 1) and diffusely infiltrating features (grade 2, grade 3, and grade 4) based on histological pathologic evaluation and genetic molecular patterns [1,3]. As the grade 4 glioma, glioblastoma multiforme (GBM) is the most aggressive type and exhibits extensive heterogeneity and poor prognosis with a high recurrence rate despite active standard treatment, including surgical resection and chemo- and radiation therapy [4].

GBM has been stratified into four subtypes, including the proneural, neuronal, mesenchymal, and classical subtypes, according to different criteria and not only gene expression signature [5]. Among the genetic alterations in GBM with different phenotypic states, a mesenchymal subtype has been associated with a worse outcome in patients with GBM [6]. In glioma tissues of GBM, mesenchymal subtype cells have the high expression of mesenchymal subtype signature genes, including chitinase 3 like 1 (*CHI3L1*), collagen type V alpha 1 chain (*COL5A1*), fibronectin 1 (*FN1*), cyclin B1 (*CCNB1*), and maternal embryonic leucine zipper kinase (*MELK*) [6,7]. Among the mesenchymal subtype signature genes, COL5A1 is a representative gene of mesenchymal subtype cell markers in glioma [8,9,10].

Glycolysis is an important metabolic pathway to produce energy for the growth and development of cancer cells, including glioma [11,12,13]. As the first key enzyme in the glycolysis pathway, hexokinase 2 (HK2) is involved in the high glycolytic activity and tumor growth in human glioma [14,15]. High glycolytic activity was found in mesenchymal GBM relative to proneural GBM [14,15,16,17].

NADPH oxidase 2 (NOX2), also known as cytochrome b-245 beta chain (CYBB), is a superoxide-generating enzyme which produces reactive oxygen species (ROS) [18,19]. NOX2 has been linked with the oncogenic signaling pathway in the regulation of the cellular metabolism and progression in cancer cells [20,21]. In addition, NOX2 is linked to epithelial-to-mesenchymal transition (EMT) and to the production of the extracellular matrix (ECM) [22,23,24]. Currently, the role of NOX2-dependent high glycolytic activity in the gain of the mesenchymal phenotype in glioma remains unclear.

Here, we showed that the levels of NOX2 were elevated in patients with GBM relative to benign glioma. NOX2 induces hexokinase 2 (HK2)-dependent high glycolytic activity in U87MG glioma cells. High levels of NOX2 are correlated with high levels of HK2 and glucose uptake in patients with GBM relative to benign glioma. Moreover, NOX2 increases the expression of mesenchymal-subtype-related genes, including *COL5A1* and *FN1* in U87MG glioma cells. High levels of NOX2 and HK2 are correlated with high levels of COL5A1 in patients with GBM relative to low-grade glioma. Our results suggest that NOX2-induced high glycolytic activity contributes to the gain of the COL5A1-mediated mesenchymal phenotype in GBM.

## 2. Materials and Methods

### 2.1. Human Study

Human subject study was conducted in accordance with the Helsinki Declaration. The protocol was approved by the Institutional Review Boards of Bundang CHA medical center (IRB number: 2021-01-024) and Soonchunhyang University Hospital Cheonan (SCHCA 2020-03-030-001). A total of 11 patients in Cohort I, which includes ganglioglioma (grade 1, n = 2), oligodendroglioma (grade 2, n = 1), diffuse astrocytoma (grade 3, n = 2), anaplastic astrocytoma (grade 3, n = 2), anaplastic oligodendroglioma (grade III, n = 1), and glioblastoma (grade 4, n = 3) (Appendix A), were obtained from Bundang CHA medical center. A total of 3 paraffin-embedded brain tissues for patients in Cohort II, which includes astrocytoma (grade 1, NBP2-77872, n = 1), astrocytoma (grade 2, NBP2-77873, n = 1), and astrocytoma (grade 3, NBP2-77874, n = 1) (Appendix A), were obtained from Novus Biologicals (Minneapolis, MN, USA).

### 2.2. ^18^F-FDG PET/CT Imaging Analysis in Human Subjects

After the patient fasted for at least 6 h, ^18^F-FDG was injected intravenously (185 MBq) into patients, and PET/CT scanning was performed 60 min later (Biograph mCT 128 scanner: Siemens Medical Solutions, Knoxville, TN, USA). Non-contrast CT images were acquired from the vertex to the skull base area for attenuation correction and lesion localization (120 kV, 120 mA, 3 mm section width, 3 mm collimation). PET images of the same area were acquired after the CT scans, in 3-dimensional mode (7 min per bed position, 21.6 cm increments). Images were reconstructed on 400 × 400 matrices using the TrueX algorithm plus time-of-flight (TOF) information (TrueX + TOF, UltraHD PET, Siemens). The images were analyzed using a dedicated workstation (Syngo.via, Siemens Medical Solutions). The volume of interest (VOI) was determined on PET/CT images or PET images co-registered with MR images to identify the tumor and contralateral normal brain tissue. The maximum tumor to background ratio (maximum TBR; the maximum activity of the tumor divided by the mean activity of the contralateral brain) was measured on ^18^F-FDG PET/CT images.

### 2.3. Immunofluorescence and Immunohistochemistry Analysis in Human Subjects

For immunofluorescence analysis, brain tissues were sectioned from paraffin embedded tissue blocks at a thickness of 4 μm. Sections were permeabilized in 0.5% Triton-X (T8787, Sigma-Aldrich, St. Louis, MO, USA), blocked in CAS-Block™ Histochemical Reagent (008120, Thermo Fisher Scientific, Waltham, MA, USA) and then stained with the following antibodies: monoclonal rabbit anti-NOX2 (1:100) (ab129068, Abcam, Cambridge, UK) in Figure 1, monoclonal mouse anti-NOX2 (1:100) (NBP1-41012, Novus, Littleton, CO, USA) in Figures 4 and 6, monoclonal rabbit anti-HK2 antibody (1:100) (#2024, Cell Signaling Technology, Danvers, MA, USA), polyclonal rabbit anti-COL5A1 antibody (1:100) (#37304, Cell Signaling Technology), polyclonal rabbit anti-GFAP (1:100) (SAB5700611, Sigma-Aldrich), and monoclonal mouse anti-GFAP (1:100) (#3670, Cell Signaling Technology). Sections were then incubated with goat anti-rabbit IgG (H+L) Alexa Fluor 488 (1:100) (A11008, Thermo Fisher Scientific) and goat anti-mouse IgG H&L Texas Red (1:100) (ab6787, Abcam) secondary antibody at 25 °C for 2 h. Fluoroshield™ with DAPI (F6057, Sigma-Aldrich) was used for nuclear staining. Stained brain sections were analyzed by THUNDER Imager Tissue (Leica Microsystems Ltd., Wetzlar, Germany). Stained brain sections were quantified by LAS X image-processing software (Leica Microsystems Ltd., Wetzlar, Germany) and ImageJ software v1.52a (Bethesda, MD, USA). In the measurement for relative intensity of fluorescence, we measured mean fluorescence intensity (MFI) for individual fluorescent channel (blue, green, or red color) in a region of interest (ROI) from images (n = 10) per individual subject using ImageJ software v1.52a (Bethesda, MD, USA). Final MFI was calculated by MFI values of an ROI compared to MFI values of Background. After we measured final MFI for each fluorescent channel separately, we quantitated the relative intensity of fluorescence in tissues of patients with glioma relative to tissues of patients with non-glioma as a basal level. To ensure objectivity, all analyses were conducted with blinded conditions by two observers who performed analyses using identical conditions per experiment. For immunohistochemistry analysis, tissues were stained with a primary antibody. The secondary antibody was biotinylated goat anti-rabbit IgG (Vector Laboratories, Burlingame, CA, USA) and biotinylated rabbit anti-goat (Vector Laboratories). Subsequently, streptavidin peroxidase complex (Vector Laboratories) was biotinylated for 2 h at 25 °C. After staining, slides were mounted by Eukitt^®^ Quick-hardening mounting medium (03989, Sigma-Aldrich). Stained sections were analyzed by Olympus BX53M microscope (Olympus, Tokyo, Japan) and quantified by the Olympus Stream software and ImageJ software v1.52a (NIH, Bethesda, MD, USA).

### 2.4. Immunofluorescence and Immunohistochemistry Analysis in Cells

Cells were plated and treated on autoclaved glass coverslips placed in 6-well cell culture plates. Cells were fixed in 4% paraformaldehyde (PFA), permeabilized in 0.5% Triton-X (T8787, Sigma-Aldrich), blocked in CAS-Block™ Histochemical Reagent (008120, Thermo Fisher Scientific), and then stained as described above.

### 2.5. Nanostring Analysis in Human Subjects

The mRNAs were analyzed using the nCounter PanCancer IO 360^TM^ Panel (NanoString nCounter Analysis System, NanoString Technologies Inc., Seattle, WA, USA). Raw gene expression values were normalized using the GeNorm algorithm nCounter Advanced Analysis (Version 2.0.115). The normalized values were transformed to log_2_ values.

### 2.6. Reagents and Antibodies

The following antibodies were used: polyclonal rabbit anti-NOX2 (ab129068, Abcam, Cambridge, UK), monoclonal mouse anti-NOX2 (NBP1-41012, Novus), monoclonal rabbit anti-HK2 antibody (#2024), monoclonal rabbit anti-LDH-A antibody (#3582, Cell Signaling Technology), monoclonal rabbit anti-LDH-A antibody (#3582, Cell Signaling Technology), polyclonal rabbit anti-COL5A1 antibody (#37304, Cell Signaling Technology), monoclonal mouse anti-GFAP (#3670, Cell Signaling Technology), monoclonal rabbit anti-GFAP (#12389, Cell Signaling Technology), and monoclonal mouse anti-β-actin (A5316, Sigma-Aldrich). Fluoroshield™ with DAPI (F6057, Sigma-Aldrich) was used for nuclear staining and mounting. Sections were mounted onto gelatin-coated slides with Canada Balsam (Wako, Tokyo, Japan) following dehydration.

### 2.7. Human Glioma Cells

Human glioma U87MG cells (ATCC^®^ HTB-14™, ATCC, Manassas, VA, USA) were used. For the experiments with U87MG cells, cells were cultured in Dulbecco’s Modified Eagle Medium (DMEM) (11995065, Thermo Fisher Scientific) containing 10% (*v*/*v*) fetal bovine serum (FBS), 100 units/mL penicillin, and 100 mg/mL streptomycin (A1261301, Thermo Fisher Scientific). The efficiency of transfection was validated in U87MG cells treated with GFP-expressing plasmid (pCMV6-AC-GFAP mammalian expression vector, Cat. No. PS100010, Origene, Rockville, MD, USA) as a positive control for transfection using lipofectamine LTX with Plus reagent (15338100, Thermo Fisher Scientific) according to the manufacturer’s instructions. The transfection reagents had no apparent effect in U87MG cells. For knockdown of human NOX2, U87MG cells were seeded (2 × 10^5^ cells/6-well cell culture plates) and transfected with MISSION^®^ esiRNA targeting human NOX2 (EHU010941, Sigma-Aldrich) against human NOX2 (NM_000397) or MISSION^®^ siRNA Universal Negative Control #1 (SIC001, Sigma-Aldrich) using lipofectamine LTX with Plus reagent (15338100, Thermo Fisher Scientific) according to the manufacturer’s instructions. Negative control had no apparent effect in U87MG cells. For overexpression of human NOX2, U87MG cells were seeded (2 × 10^5^ cells/6-well cell culture plates) and transfected with pCMV6-AC-GFP constructs containing human NOX2 (NM_000397) (RG207544, Origene, Rockville, MD, USA) or pCMV6-AC-GFP vector (PS100010, Origene) using lipofectamine LTX with Plus reagent (15338100, Thermo Fisher Scientific) according to the manufacturer’s instructions. Negative control had no apparent effect in U87MG cells. Cellular morphology was analyzed by EVOS M5000 Imaging System (Thermo Fisher Scientific) according to the manufacturer’s instructions.

### 2.8. Quantification of Gene Expression

The quantification of gene expression levels was performed by QuantStudio™ 5 Real-Time PCR System (Thermo Fisher Scientific). A 10 μL mixture containing 2 μL of DNA template and a set of gene-specific primers was mixed with 10 μL of 2 × SYBR Green PCR Master Mix (4309155, Applied Biosystems, Waltham, MA, USA) and then subjected to real time-polymerase chain reaction (RT-PCR) quantification. The primer sequences were used as follows: human *COL5A1*, forward 5′-GATTGAGCAGATGAAACGGCC-3′, reverse 5′-CCTTGGTTAGGATCGACCCAG-3′; human *FN1*, forward 5′-TGGTGGCCACTAAATACGAA-3′, reverse 5′- GGAGGGCTAACATTCTCCAG-3′; human *GAPDH*, forward 5′-CAACAGCGACACCCACTCCT-3′, reverse 5′-CACCCTGTTGCTGTAGCCAAA-3′. *GAPDH* was used as a loading control. The RT-PCR assay was carried out as follows: a denaturation step for 10 min at 95 °C; followed by 40 cycles of 30 s at 95 °C, 30 s at 60 °C, and 30 s at 72 °C.

### 2.9. Immunoblot Analysis

Cells were harvested and lysed in NP40 Cell Lysis Buffer (FNN0021, Thermo Fisher Scientific, Waltham, MA, USA). Lysates were centrifuged at 15,300× *g* for 10 min at 4 °C, and the supernatants were obtained. The protein concentrations were determined by using the Bradford assay kit (500-0006, Bio-Rad Laboratories, Hercules, CA, USA). Proteins were electrophoresed on NuPAGE 4–12% Bis-Tris gels (Thermo Fisher Scientific) and transferred to Protran nitrocellulose membranes (10600001, GE Healthcare Life science, Pittsburgh, PA, USA). Membranes were blocked in 5% (*w*/*v*) bovine serum albumin (BSA) (9048-46-8, Santa Cruz Biotechnology, Dallas, TX, USA) in TBS-T (TBS (170-6435, Bio-Rad Laboratories, Hercules, CA, USA) and 1% (*v*/*v*) Tween-20 (170-6531, Bio-Rad Laboratories) for 30 min at 25 °C. Membranes were incubated with primary antibody (1:1000) diluted in 1% (*w*/*v*) BSA in TBS-T for 16 h at 4 °C and then with the horseradish peroxidase (HRP)-conjugated secondary antibody (goat anti–rabbit IgG–HRP (sc-2004) (1:2500) and goat anti–mouse IgG–HRP (sc-2005) (1:2500) from Santa Cruz Biotechnology) diluted in TBS-T for 0.5 h at 25 °C. Immunoreactive bands were detected with the SuperSignal West Pico Chemiluminescent Substrate (34078, Thermo Scientific). All the whole western blot figures can be found in the Appendix A.

### 2.10. Glycolysis Activity Assay

For the glycolytic function assay, human glioma cells (5 × 10^4^ cells/well) were plated on XF96 cell culture microplates (101085-004, Agilent Technologies, Inc., Santa Clara, CA, USA). The levels of ECAR, which is a parameter of glycolytic flux and activity, were measured by the Seahorse XF96^e^ bioanalyzer and the XF Glycolysis Stress Test Kit (102194-100, Agilent Technologies, Inc.) according to the manufacturer’s instructions. The levels of ECAR were measured in cells that were treated with glucose (10 mM), oligomycin (2 μM), and 2-deoxyglucose (2DG) (10 mM).

### 2.11. Analysis of Datasets

We downloaded RNA-seq and survival data of lower grade glioma (LGG) and glioblastoma (GBM) from the Xena TCGA database (https://xenabrowser.net/ accessed date: from 10 November 2020 to 15 February 2021). In total, 693 patients participated in this study. Among the data, there were grade 2 (n = 257), grade 3 (n = 270) and grade 4 (n = 166) patients. Expression values of genes were transformed to a 0–1 standard scale and merged with survival and pathological grade data. To evaluate results from TCGA data, we used the REMBRANDT dataset from GlioVis, which is a web-based tool for visualizing statistical analyses of various glioma studies (http://glio\vis.bioinfo.cnio.es (accessed on 12 December 2021)) [25]. A total of 313 patients (grade 2 (n = 98), grade 3 (n = 85), and grade 4 (n = 130)) were included in the REMBRANDT data. The survival data showed overall survival time and events. This process was performed with Python 2.7.

### 2.12. Statistical Analysis

All statistical analyses were performed using the statistical software package GraphPad Prism version 8.0 (GraphPad Software Inc., San Diego, CA, USA). The two-tailed Student’s t-test was used for two-group comparison, and the analysis of variance (ANOVA) (with post hoc comparisons using Dunnett’s test) was used for multi-group comparison. To evaluate statistical differences of gene expression between each neoplasm histologic grade, the *t*-test or Wilcoxon test was performed with respect to the TCGA and NanoString datasets, respectively. Pearson’s correlation analysis was used for evaluating associations between the expression of two genes. Linear regression was performed to confirm the relationship between the expression of two genes. Kaplan–Meier estimation and Cox-regression analysis were performed to analyze the prognosis of genes with overall survival data. Concordance index was calculated to provide accuracy with survival proportion. Kaplan–Meier estimation was performed to analyze prognosis of genes with overall survival data. All statistical analyses were performed using Rstudio (Version: 1.1.456). *p* values lower than 0.05 (*, *p* < 0.05, **, *p* < 0.01, ***, *p* < 0.001) were considered statistically significant.

## 3. Results

### 3.1. The Levels of NOX2 Are Elevated in Patients with GBM Than Low-Grade Glioma

To investigate the role of NOX2 in the progression of GBM, we analyzed whether the levels of NOX2 were elevated in glioma tissues from patients with GBM (Appendix A). We measured the protein levels of NOX2 in glial fibrillary acidic protein (GFAP)-positive glioma cells in patients with high-grade glioma (grade 4 (G4), GBM (n = 3), and grade 3 (G3), anaplastic astrocytoma (n = 3)) and low-grade glioma (grade 1 (G1), ganglioglioma (n = 2), and grade 2 (G2), diffuse astrocytoma (n = 3)) using immunofluorescence staining (Figure 1a and Appendix A). Since GFAP is highly expressed in the glioma cells of astrocytomas and glioblastomas [26], we used GFAP for a marker of glioma cells in glioma tissues from patients with GBM. The intensity of NOX2-positive staining in GFAP-positive glioma cells was significantly elevated in patients with GBM (G4) relative to G1–G3 glioma. In addition, the intensity of NOX2-positive staining was higher in GBM (G4) and G3 glioma than that in low-grade glioma (G1, G2) (Figure 1a,b). The number of glioma cells that have a subcellular co-localization of NOX2 and GFAP was significantly increased in patients with GBM (G4) relative to that in low-grade glioma (G1, G2) (Figure 1c). Consistently, the levels of the NOX2 gene were significantly elevated in GBM (G4) compared to low-grade glioma (G2) in the analysis of the GBM and LGG datasets from TCGA (Figure 1d) and REMBRANDT (Appendix A). Moreover, the patients with high levels of NOX2 gene expression (median survival: 1062 days) had poor survival rates compared to patients with low levels of the NOX2 gene (median survival: 2282 days) in the analysis of the GBM and LGG datasets from TCGA (Figure 1e). These results suggest that the levels of NOX2 are elevated in patients with GBM compared to low-grade glioma.

### 3.2. NOX2 Induces the Activation of HK2-Dependent Glycolysis in Human Glioma Cells

Next, we investigated whether NOX2 could affect high glycolytic activity in human glioma cells. We analyzed the effects of NOX2 knockdown in the activation of HK2-dependent glycolysis in U87MG glioma cells (Figure 2a). We measured the extracellular acidification rate (ECAR) as a surrogate of glycolysis activity by the quantification of lactate production. The glycolytic activity was measured by the sequential addition of glucose, the substrate of glycolysis; oligomycin, a selective inhibitor of mitochondrial respiration; and 2-deoxyglucose (2-DG), a specific inhibitor of glycolysis (Figure 2a). NOX2 knockdown significantly suppressed the levels of ECAR in response to glucose relative to the control (Figure 2a,b). Next, we examined whether NOX2 could regulate the expression of HK2, the first key enzyme in the glycolysis pathway, in U87MG glioma cells. NOX2 knockdown reduced the protein levels of HK2 compared to the control, whereas the protein levels of lactate dehydrogenase-A (LDH-A), an enzyme for the final step of glycolysis, were not changed (Figure 2c,d). Consistently, the over-expression of NOX2 resulted in the higher ECAR levels in response to glucose than that in the control (Figure 2e,f). The over-expression of NOX2 increased the protein levels of HK2 relative to the control, whereas the protein levels of LDH-A were comparable (Figure 2g,h). Moreover, the inhibition of NOX2 activity by GSK2795039, a selective NOX2 inhibitor, significantly suppressed the levels of ECAR in response to glucose and the protein levels of HK2 relative to the control (Figure 3a,b). GSK2795039 decreased the protein levels of HK2 relative to the control, whereas the protein levels of LDH-A were comparable (Figure 3c,d). These results suggest that NOX2 induces the activation of HK2-dependent glycolysis in human glioma cells.

### 3.3. High Levels of NOX2 Contribute to High Levels of Glucose Uptake and HK2 Expression in Patients with GBM

Next, we investigated whether high levels of NOX2 contribute to high levels of glucose uptake and HK2 expression in patients with GBM. First, we analyzed the levels of glucose uptake in patients with glioma using ^18^F-FDG PET with MRI analysis. The high radiomics signature of glucose uptake and the levels of maximum TBR in the ^18^F-FDG PET/MRI images were significantly elevated in patients with GBM relative to those in G1–G3 glioma patients (Figure 4a,b). We analyzed the protein levels of HK2 in NOX2-positive glioma cells of patients with GBM using immunofluorescence staining (Figure 4c and Appendix A). The intensity of HK2-positive staining and the number of cells that have subcellular co-localization of HK2 and NOX2 were elevated in patients with GBM (G4) and G3 glioma relative to those in patients with low-grade glioma (G1, G2) (Figure 4d,e). Notably, the levels of HK2 and the number of cells that have subcellular co-localization of HK2 and NOX2 were significantly increased in patients with GBM relative to those in G2 and G3 gliomas (G2, G3) (Figure 4d,e). Similar to the cohort I, the levels of HK2 and NOX2 were elevated in patients with G3 glioma (G3) relative to those in G1 and G2 gliomas (G1, G2) in the cohort II (Appendix A). The intensity of HK2-positive staining in NOX2-positive staining (Appendix A) and the number of cells showing subcellular co-localization of HK2 and NOX2 were increased in patients with G3 glioma (G3) relative to those in G1 and G2 gliomas (G1, G2) (Appendix A). Moreover, the levels of the HK2 gene were significantly elevated in GBM (G4) compared to G2 glioma (G2) in the analysis of the GBM and LGG datasets from TCGA (Figure 4f) and REMBRANDT (Appendix A). The expression levels of HK2 and NOX2 mRNA were positively correlated in patients with glioma in the analysis of the GBM and LGG datasets from TCGA (Figure 4g). The high levels of the HK2 gene were associated with the poor survival of patients with glioma in the analysis of the GBM and LGG datasets from TCGA (Figure 4h). These results suggest that high levels of NOX2 contribute to high levels of glucose uptake and HK2 expression in patients with GBM.

### 3.4. NOX2 and HK2 Induces COL5A1-Mediated Mesenchymal Phenotype in Human Glioma Cells

Next, we investigated whether NOX2-dependent high glycolytic activity could affect the gain of the COL5A1-mediated mesenchymal phenotype in human glioma cells. First, we analyzed the effects of NOX2 knockdown on the change of COL5A1 gene expression in U87MG glioma cells. NOX2 knockdown by NOX2 siRNA suppressed the levels of COL5A1 gene expression compared to that in control (Figure 5a). In addition, the levels of FN1 expression were reduced by NOX2 knockdown (Figure 5a). Similary, NOX2 knockdown suppressed the protein levels of COL5A1 and FN1 relative to that in control (Figure 5b). Moreover, the over-expression of HK2 significantly increased the gene and protein levels of COL5A1 and FN1 compared to that in control (Figure 5c,d). Consistently, the over-expression of NOX2 induced the morphological features of mesenchymal cells, including spindle-shape and large flattened-shape relative to that in control (Figure 5e). The length of the spindle-shape in cells was increased by NOX2 over-expression compared to that in control (Figure 5e). Similar with NOX2, the over-expression of HK2 increased the gene and protein levels of COL5A1 and FN1 compared to that in the control (Figure 5f,g). The over-expression of HK2 induced the morphological features of mesenchymal cells, including spindle-shape and large flattened-shape relative to that in the control (Figure 5h). These results suggest that NOX2 and HK2 induce COL5A1-mediated mesenchymal phenotype in human glioma cells.

### 3.5. NOX2 and HK2 Contributes to the Gain of COL5A1-Mediated Mesenchymal Phenotype in Patients with GBM

Next, we investigated whether NOX2 could affect the gain of COL5A1-mediated mesenchymal phenotype in GBM. First, we identified the levels of mesenchymal phenotype by gene expression profile using Nanostring analysis in glioma tissues of patients with GBM (Appendix A). The levels of mesenchymal subtype signature genes were elevated in glioma tissues of patients with GBM (G4) relative to that in patients with G2 and G3 gliomas (G2, G3) (Appendix A). Next, we analyzed the expression levels of COL5A1 in NOX2-positive cells of glioma tissues from patients with GBM (Figure 6a and Appendix A). The intensity of COL5A1-positive staining was increased in patients with GBM (G4) and G3 glioma (G3) compared to that in patients with G1 and G2 gliomas (G1, G2) (Figure 6b). The number of cells showing subcellular co-localization of COL5A1 and NOX2 was significantly increased in patients with GBM (G4) and G3 glioma (G3) compared to that in G1 and G2 glioma (Figure 6c). Similarly, the number of cells showing subcellular co-localization of COL5A1 and HK2 was increased in patients with GBM (G4) and G3 glioma compared to that in G1 and G2 glioma (Figure 6d and Appendix A). Notably, the intensity of COL5A1-positive staining and the number of cells showing subcellular co-localization of COL5A1 and NOX2 or HK2 were significantly elevated in in patients with GBM (G4) relative to that in patients with G3 glioma (G3) (Figure 6c–e). Furthermore, the expression of the COL5A1 gene was positively correlated with the expression of the NOX2 and HK2 genes in the analysis of the GBM and LGG datasets from TCGA (Figure 6f,g) and REMBRANDT (Appendix A). These results suggest that NOX2 and HK2 contribute to the gain of the COL5A1-mediated mesenchymal phenotype in patients with GBM.

## 4. Discussion

Here we demonstrate that NOX2-induced high glycolytic activity contributes to the gain of the COL5A1-mediated mesenchymal phenotype in GBM. Our results suggest that NOX2 induces HK2-dependent high glycolytic activity in GBM. In addition, we suggest that NOX2-induced high glycolytic activity increases the gain of the COL5A1-mediated mesenchymal phenotype in GBM. Our findings provide a molecular mechanism by which NOX2-dependent glycolytic activity contributes to the elevation of the mesenchymal phenotype in GBM.

Glucose metabolism is an important metabolic pathway in the development and progression of GBM [27,28,29]. Glioma cells show rapid glucose uptake from the microenvironment and accelerated glycolysis [30,31]. The EGFR-driven signaling pathway induces not only the uptake and utilization of glucose but also the uptake and utilization of acetate in glioma cells [32]. MYC promotes both glycolytic flux and glutamine levels in glioma cells of GBM [33,34,35]. In our findings, we suggest that NOX2 could be a critical molecule for high glycolytic activity in glioma cells of GBM. Since our results show that the high levels of NOX2 contribute to the high glucose uptake in patients with GBM, we suggest that the elevation of NOX2-dependent high glycolysis might be a critical mechanism for high glycolytic phenotype of GBM. Although our results suggest that the role of NOX2 in the regulation of glucose metabolism, further study is needed in the changes of cellular proliferation and growth by NOX2 in glioma cells.

The mesenchymal subtype of GBM is associated with resistance to various chemotherapeutic agents as well as to radiotherapy, which inevitably leads to recurrence after surgical resection [36,37,38]. As a representative gene of mesenchymal subtype of GBM, *COL5A1* induces the malignancy of GBM via the activation of migration/invasion ability in GBM cells [39]. In addition, the elevation of COL5A1 correlates with poor survival in patients with glioma [39]. Our results showed that NOX2 promotes the elevation of COL5A1 expression in human glioma cells. In addition, our results showed that high levels of NOX2 correlate with high levels of COL5A1 in GBM cells of patients with GBM. Our findings suggest that NOX2 might be critical for the regulation of COL5A1-mediated mesenchymal phenotype in GBM cells. In addition, our findings suggest that NOX2 might be a new target for controlling GBM invasiveness. Although the correlation coefficient of NOX2 with COL5A1 or HK2 with COL5A1 is weak in our results, we observed consistent results for correlations including NOX2 and HK2 with COL5A1 with the REMBRANDT dataset presented. In our results, we showed the role of NOX2 in the gain of the COL5A1-mediated mesenchymal phenotype in GBM. Further study is needed in the regulation of other mesenchymal subtype signature genes and tumor microenvironment by NOX2 during the mesenchymal transition in GBM. In addition, further study is needed to investigate the effects of chemotherapeutic agents and the changes of genes related to epithelial—mesenchymal transition in NOX2 overexpressing glioma cells.

While high glycolytic phenotype is a metabolic feature of GBM, the underlying molecular mechanism by which the high glycolytic phenotype regulates the gain of the mesenchymal subtype in GBM is still unclear. In our current study, we suggest that the activation of NOX2-dependent high glycolytic activity could be a critical pathway for the gain of the COL5A1-mediated mesenchymal phenotype in GBM. Since our results suggest that NOX2 contributes to the gain of the mesenchymal subtype in GBM, NOX2 might be a molecular target of therapeutic approach for inhibition of mesenchymal phenotype in GBM. We anticipate that the inhibition of NOX2 could be useful in the treatment of patients with GBM in the future. Although we found the role of NOX2 in the gain of the mesenchymal phenotype in GBM, our current study has some limitations. First, we did not show correlations of NOX2 with prognostic and predictive markers affecting clinical outcome and other potential confounding factors. Second, an appropriate sample size is critical for studies that detect clinically relevant differences. Our collected cohort has a small sample size that includes various glioma grades. Due to the locational characteristics of the disease, it was difficult to obtain a normal control group, so it was replaced with grade 1 ganglioglioma. On the other hand, the open cohorts used include a large population of patients. In general, small sample sizes do not provide sufficient statistical power and result in type II error, which can ultimately lead to erroneous conclusions. Larger sample sizes make the statistic much more analytic, but it can sometimes be a hurdle. If the sample size is too large, first, a lot of financial and human resources are required, and results with exaggerated tendencies may be reached by emphasizing the statistical differences of clinically irrelevant influences. To provide higher clinical value, we also presented the results of additional open cohorts, but these limitations require the careful interpretation of our results. Therefore, we need to overcome these limitations and further study the correlation of NOX2 with clinical outcomes and other potential confounding factors, extending to a diverse glioma cohort with more appropriate sample sizes to overcome these limitations and identify clearer impacts.

In conclusion, our findings demonstrate that the metabolic function of NOX2 could be critical for the development of the COL5A1-mediated mesenchymal subtype in GBM.

## Figures and Tables

**Figure 1 cancers-14-00516-f001:**
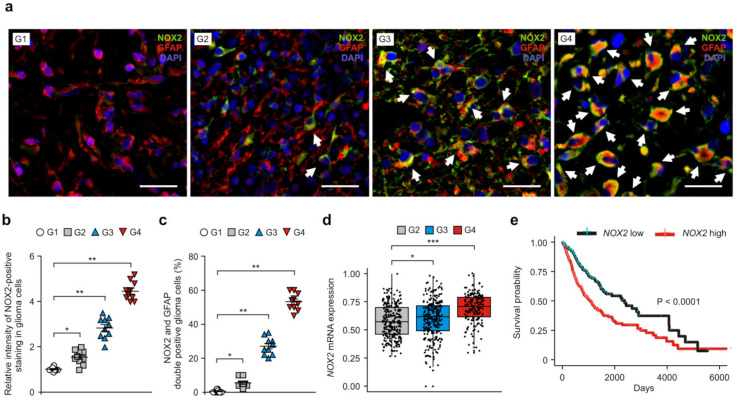
The levels of NOX2 are elevated in patients with GBM compared to low-grade glioma. (**a**) Representative immunofluorescence images of NOX2 staining in patients with GBM (G4), G3 glioma (G3), G2 glioma (G2), and G1 glioma (G1) showing NOX2 (green), GFAP (red, glioma cells), and DAPI-stained nuclei (blue) (n = 3 per group, n = 10 images per individual subject). Scale bars, 20 μm. White arrows indicate NOX2 and GFAP-positive glioma cells. (**b**,**c**) Relative intensity of NOX2-positive staining (**b**) and quantification of NOX2 and GFAP-positive glioma cells (**c**) from (**a**). (**d**) The levels of NOX2 mRNA in patients with GBM and LGG dataset from TCGA. Wilcoxon test with two-sided and one-sided. (**e**) The survival curve of patients with low (black) (n = 347) and high levels (red) (n = 346) of NOX2 mRNA (n = 693) from GBM and LGG datasets from TCGA. The two groups were divided according to median value. *p* < 0.0001; Kaplan–Meier estimation test with two-sided and one-sided. The results are representative of three independent experiments and are shown as the mean ± SEM. *** *p* < 0.001, ** *p* < 0.01, * *p* < 0.05; ANOVA or Student’s two-tailed *t*-test.

**Figure 2 cancers-14-00516-f002:**
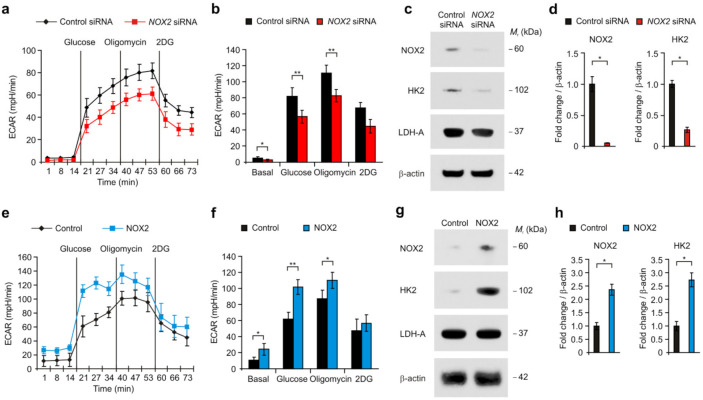
NOX2 induces the activation of HK2-dependent glycolysis in human glioma cells. (**a**) The levels of extracellular acidification rate (ECAR) for glycolysis of glucose and (**b**) quantification of ECAR levels in human glioma cells treated with *NOX2* siRNA or control siRNA. (**c**) Representative immunoblot images and (**d**) quantification of NOX2, HK2, and LDH-A protein levels in human glioma cells treated with *NOX2* siRNA or control siRNA. β-actin was used as a loading control. (**e**) The levels of extracellular acidification rate (ECAR) for glycolysis of glucose and (**f**) quantification of ECAR levels in human glioma cells with NOX2 over-expression or control. (**g**) Representative immunoblot images and (**h**) quantification of NOX2, HK2, and LDH-A protein levels in human glioma cells with NOX2 over-expression or control. Β-actin was used as a loading control. The results are representative of three independent experiments and are shown as the mean ± SEM. ** *p* < 0.01, * *p* < 0.05; ANOVA or Student’s two-tailed *t*-test.

**Figure 3 cancers-14-00516-f003:**
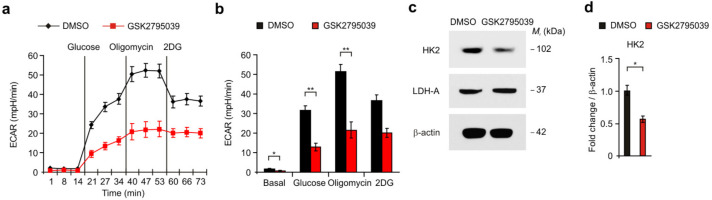
Inhibition of NOX2 suppresses the activation of HK2-dependent glycolysis in human glioma cells. (**a**) The levels of extracellular acidification rate (ECAR) for glycolysis of glucose and (**b**) quantification of ECAR levels in human glioma cells treated with GSK2795039 or DMSO. (**c**) Representative immunoblot images and (**d**) quantification of NOX2, HK2, and LDH-A protein levels in human glioma cells treated with GSK2795039 or DMSO. β-actin was used as a loading control. The results are representative of three independent experiments and are shown as the mean ± SEM. ** *p* < 0.01, * *p* < 0.05; ANOVA or Student’s two-tailed *t*-test.

**Figure 4 cancers-14-00516-f004:**
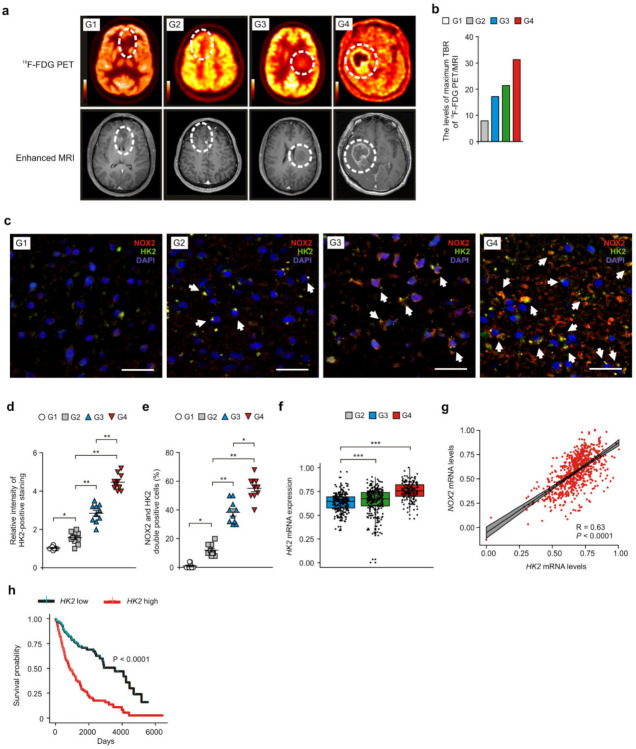
High levels of NOX2 contribute to high levels of glucose uptake and HK2 expression in patients with GBM. (**a**,**b**) Representative ^18^F-FDG PET /MRI images (**a**) and quantification of maximum TBR levels for ^18^F-FDG uptake (**b**) in brains from patients with GBM (G4), G3 glioma (G3), G2 glioma (G2), and G1 glioma (G1). (**c**) Representative immunofluorescence images of NOX2 and HK2 staining in patients with GBM (G4), G3 glioma (G3), G2 glioma (G2), and G1 glioma (G1) showing NOX2 (red), HK2 (green), and DAPI-stained nuclei (blue) (n = 3 per group, n = 10 images per individual subject). Scale bars, 20 μm. White arrows indicate NOX2 and HK2-positive cells. (**d**,**e**) Relative intensity of HK2-positive staining (**d**) and quantification of NOX2 and HK2-positive glioma cells (**e**) from (**c**). (**f**) The levels of HK2 mRNA in patients with GBM and LGG datasets from TCGA. Wilcoxon test with two-sided and one-sided. (**g**) Spearman’s correlation coefficient analysis between NOX2 and HK2 genes in patients (n = 693) with GBM and LGG dataset from TCGA. R = 0.63, *p* < 0.0001. (**h**) The survival curve of patients with low (black) (n = 347) and high levels (red) (n = 346) of HK2 mRNA (n = 693) from GBM and LGG dataset from TCGA. The two groups were divided according to median value. *p* < 0.0001; Kaplan–Meier estimation test with two-sided and one-sided. The results are representative of three independent experiments and are shown as the mean ± SEM. *** *p* < 0.001, ** *p* < 0.01, * *p* < 0.05; ANOVA or Student’s two-tailed *t*-test.

**Figure 5 cancers-14-00516-f005:**
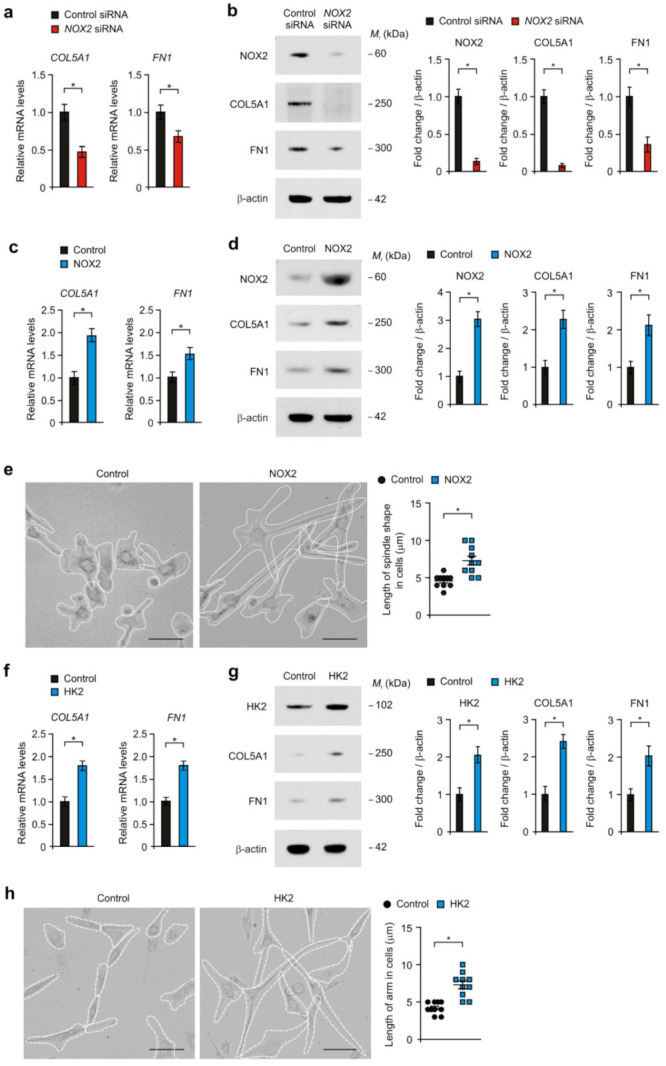
NOX2 and HK2 induce COL5A1-mediated mesenchymal phenotype in human glioma cells. (**a**) The levels of *COL5A1* and *FN1* mRNA and (**b**) representative immunoblot images (left) and quantification (right) of COL5A1 and FN1 protein levels in human glioma cells treated with *NOX2* siRNA or control siRNA. β-actin was used as a loading control. (**c**) The levels of *COL5A1* and *FN1* mRNA and (**d**) representative immunoblot images (left) and quantification (right) of COL5A1 and FN1 protein levels in human glioma cells with NOX2 over-expression (NOX2) or control (Control). β-actin was used as a loading control. (**e**) Representative images of cellular morphology in U87MG glioma cells with NOX2 over-expression (HK2) or control (Control) (left). Quantification of spindle shape length in U87MG glioma cells with NOX2 over-expression or control (right). Scale bars, 20 μm. (**f**) The levels of *COL5A1* and *FN1* mRNA and (**g**) representative immunoblot images (left) and quantification (right) of COL5A1 and FN1 protein levels in U87MG glioma cells with HK2 over-expression (HK2) or control (Control). (**h**) Representative images of cellular morphology in U87MG glioma cells with HK2 over-expression (HK2) or control (Control) (left). Scale bars, 20 μm. Quantification of spindle shape length in human glioma cells with HK2 over-expression (HK2) or control (Control) (right). The results are representative of three independent experiments and are shown as the mean ± SEM. * *p* < 0.05; Student’s two-tailed *t*-test.

**Figure 6 cancers-14-00516-f006:**
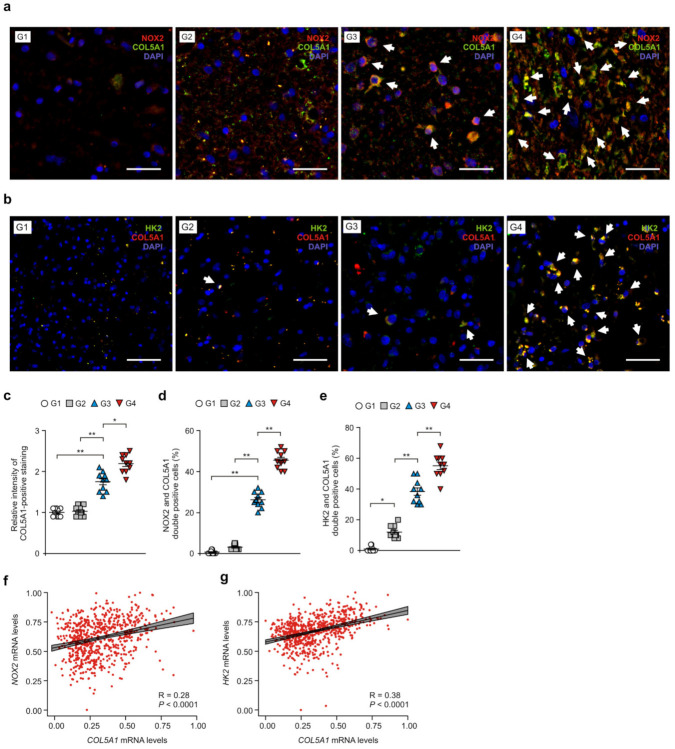
NOX2 and HK2 contributes to the gain of the COL5A1-mediated mesenchymal phenotype in patients with GBM. (**a**) Representative immunofluorescence images of NOX2 and COL5A1 staining in patients with GBM (G4), G3 glioma (G3), G2 glioma (G2), and G1 glioma (G1) showing NOX2 (red), COL5A1 (green), and DAPI-stained nuclei (blue) (n = 3 per group, n = 10 images per individual subject). Scale bars, 20 μm. White arrows indicate NOX2 and COL5A1-positive cells. (**b**) Representative immunofluorescence images of HK2 and COL5A1 staining in patients with GBM (G4), G3 glioma (G3), G2 glioma (G2), and G1 glioma (G1) showing COL5A1 (red), HK2 (green), and DAPI-stained nuclei (blue) (n = 3 per group, n = 10 images per individual subject). Scale bars, 20 μm. White arrows indicate HK2 and COL5A1-positive cells. (**c**,**d**) Relative intensity of COL5A1-positive staining (**c**) and quantification of NOX2 and COL5A1-positive glioma cells (**d**) from (**a**). (**e**) Quantification of HK2 and COL5A1-positive glioma cells from (**b**). (**f**) A Spearman’s correlation coefficient analysis between HK2 and COL5A1 gene patients (n = 693) with GBM and LGG datasets from TCGA. R = 0.28, *p* < 0.0001. (**g**) Spearman’s correlation coefficient analysis between *HK2* and *COL5A1* gene patients (n = 693) with GBM and LGG datasets from TCGA. R = 0.38, *p* < 0.0001. The results are representative of three independent experiments and are shown as the mean ± SEM. ** *p* < 0.01, * *p* < 0.05; ANOVA or Student’s two-tailed *t*-test.

## Data Availability

The data presented in this study are available on request from the corresponding author.

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
