# Peer review of "NOX2-Induced High Glycolytic Activity Contributes to the Gain of COL5A1-Mediated Mesenchymal Phenotype in GBM"

_cancers, 2022, doi:10.3390/cancers14030516_

Round 1

Reviewer 1 Report

While the authors have revised aspects of the manuscript, my major concerns remain as the methodological problems with data generation and presentation are remaining.

Author Response

Response to Cancers Reviewer 1 Comments

Comment

While the authors have revised aspects of the manuscript, my major concerns remain as the methodological problems with data generation and presentation are remaining.

Response:

We agree with your concerns for methodological problems with data generation and presentation. As your concerns, our current study still has some limitations. Therefore, we added the below description for our limitation of data generation and presentation as your aspects in Discussion section at Line 505-526. We hope to perform our further study supporting our current findings following your suggestion.

‘First, we did not show correlations of NOX2 with prognostic and predictive markers affecting clinical outcome and other potential confounding factors. Second, an appropriate sample size is critical for studies that detect clinically relevant differences. Our collected cohort has a small sample size that includes various glioma grades. Due to the locational characteristics of the disease, it was difficult to obtain a normal control group, so it was replaced with grade 1 ganglioglioma. On the other hand, the open cohorts used include a large population of patients. In general, small sample sizes do not provide sufficient statistical power and result in type II error, which can ultimately lead to erroneous conclusions. Larger sample sizes make the statistic much more analytic, but it can sometimes be a hurdle. If the sample size is too large, first, a lot of financial and human resources are required, and results with exaggerated tendencies may be reached by emphasizing statistical differences of clinically irrelevant influences. To provide higher clinical value, we also presented the results of additional open cohorts, but these limitations require careful interpretation of our results. Therefore, we need to overcome these limitations and further study the correlation of NOX2 with clinical outcomes and other potential confounding factors, extending to a diverse glioma cohort with more appropriate sample sizes to overcome these limitations and identify clearer impacts.’

Reviewer 2 Report

The original research article entitled “NOX2-induced high glycolytic activity contributes to the gain of COL5A1-mediated mesenchymal phenotype in GBM” by Park et al., showed the mechanistic gain of mesenchymal phenotype in GBM. Authors showed elevated NOX2 levels GBM patients and increased glycolytic activity in glioma cells due to high HK2 induced by NOX2. Further, authors showed that the NOX2-induced high glycolytic activity contributes to the gain of COL5A1-mediated mesenchymal phenotype in GBM. Finally, authors concluded that the metabolic function of NOX2 could be critical for the development of COL5A1-mediated mesenchymal subtype in GBM.

All the experiments were elegantly performed and the results supports authors’ conclusions.  

Author Response

Response to Cancers Reviewer 2 Comments

Comment

The original research article entitled “NOX2-induced high glycolytic activity contributes to the gain of COL5A1-mediated mesenchymal phenotype in GBM” by Park et al., showed the mechanistic gain of mesenchymal phenotype in GBM. Authors showed elevated NOX2 levels GBM patients and increased glycolytic activity in glioma cells due to high HK2 induced by NOX2. Further, authors showed that the NOX2-induced high glycolytic activity contributes to the gain of COL5A1-mediated mesenchymal phenotype in GBM. Finally, authors concluded that the metabolic function of NOX2 could be critical for the development of COL5A1-mediated mesenchymal subtype in GBM.

All the experiments were elegantly performed and the results supports authors’ conclusions.

Response:

Thank you for all your comments. They were valuable for our paper.

Reviewer 3 Report

The manuscript is well designed and aims to investigate how NOX2-induced high glycolytic activity is contributed to the mesenchymal phenotype in GBM. The knowledge gained in this manuscript ultimately improves our understanding of critical metabolic pathways leading to the mesenchymal subtype in GBM. The provided clinical data is well engaged with translational research and supports the main manuscript conclusion. I recommend the manuscript for publication in Cancers with the following suggestions:

  1. NOX2 has an official gene symbol CYBB. Please include this information in the abbreviation list or mention this somewhere in the text for a broad range of readers.
  2. Line 301, Figure 1e. I assume that the survival curves are provided for CYBB (NOX2); please confirm this. Please specify the number of patients with low (black) (n=?) and high level (red) (n=?) of NOX2 mRNA (n=693) from the GBM and LGG dataset for survival curves.  What is the cutoff? (Is this the mean value?) Please include this information.
  3. Line 382, Figure 4g.Please provide a description and number of patients in the subset chosen to evaluate the correlation between NOX2 and HK2 genes.
  4. Line 383, Figure 4h. Please specify the number of patients with low (black) (n=?) and high level (red) (n=?) of HK2 mRNA (n=693) from the GBM and LGG dataset for survival curves. What is the cutoff?
  5. Line 451, Figures 6f and 6g. Please provide a description and number of patients in the subsets chosen to evaluate the correlation between NOX2/COL5A1 and HK2/COL5A1 genes.  
  6. Please be consistent with the capitalization of Figure abbreviations such as either a capital letter (E) or lowercase (e).

Author Response

Response to Cancers Reviewer 3 Comments

Comments
The manuscript is well designed and aims to investigate how NOX2-induced high glycolytic activity is contributed to the mesenchymal phenotype in GBM. The knowledge gained in this manuscript ultimately improves our understanding of critical metabolic pathways leading to the mesenchymal subtype in GBM. The provided clinical data is well engaged with translational research and supports the main manuscript conclusion. I recommend the manuscript for publication in Cancers with the following suggestions:

  1. NOX2 has an official gene symbol CYBB. Please include this information in the abbreviation list or mention this somewhere in the text for a broad range of readers.

Response: As your comment, we provided the information for CYBB in Introduction section at Line 73.

  1. Line 301, Figure 1e. I assume that the survival curves are provided for CYBB (NOX2); please confirm this. Please specify the number of patients with low (black) (n=?) and high level (red) (n=?) of NOX2 mRNA (n=693) from the GBM and LGG dataset for survival curves.  What is the cutoff? (Is this the mean value?) Please include this information.

Response: As your comment, we specified the number of patients with low (black) (n=347) and high level (red) (n=346) of NOX2 mRNA at Line 301. We provided the information of cutoff as ‘The two groups were divided according to median value’ at Line 303.

  1. Line 382, Figure 4g. Please provide a description and number of patients in the subset chosen to evaluate the correlation between NOX2 and HK2 genes.

Response: As your comment, we provided a description and number of patients in the subset chosen to evaluate the correlation between NOX2 and HK2 genes at Line 384. The number of patients for evaluating the correlation between NOX2 and HK2 genes was 693.

  1. Line 383, Figure 4h. Please specify the number of patients with low (black) (n=?) and high level (red) (n=?) of HK2 mRNA (n=693) from the GBM and LGG dataset for survival curves. What is the cutoff?

Response: As your comment, we specified the number of patients with low (black) (n=347) and high level (red) (n=346) of NOX2 mRNA at Line 385. We provided the information of cutoff as ‘The two groups were divided according to median value’ at Line 386.

  1. Line 451, Figures 6f and 6g. Please provide a description and number of patients in the subsets chosen to evaluate the correlation between NOX2/COL5A1 and HK2/COL5A1 genes.  

Response: As your comment, we provided a description and number of patients in the subset chosen to evaluate the correlation between NOX2/COL5A1 and HK2/COL5A1 genes at Line 457-458. The number of patients for evaluating the correlation between NOX2 and HK2 genes was 693.

  1. Please be consistent with the capitalization of Figure abbreviations such as either a capital letter (E) or lowercase (e).

Response: As your comment, we revised the capitalization of figure abbreviations using lowercase.

Round 2

Reviewer 1 Report

I thank the authors for clarifying their points.

This manuscript is a resubmission of an earlier submission. The following is a list of the peer review reports and author responses from that submission.

Round 1

Reviewer 1 Report

Very interesting article. Some aspects need to be addressed before the manuscript can be considered for publication:

-Fig 1e: please report median survival for each group

-Fig 1a-b: You report relative intensity. This is not a standard measure. Please provide a detailed methodological section on how you assessed and calculated relativ intensity.

-Fig 1d: The larger the dataset, the "easier" you may observe significant differences. With a dataset of > 600 patients, the power of the dataset results in significant differences even at small absolute differences. Please present values to the reader to better assess whether the observed statistical difference is clinically signifiant (which does not have to be the case).

-All above remarks may be addressed to figure 4.

-Why did you choose 690 patients? Did you perform an a priori power calculation? If not, please state in your methods section

-Again, relative intensity is used in Figure 6

-By definition, Spearman's R of <0.4 ist only a "weak" correlation (with only one correlation "very weak" being poorer). Hence, The findings of your analysis in figure 6f (and g) are definitely to be discussed critically. Further, such observation needs to be critically discussed.

-You use data from a large cohort of patients including various glioma grades (1-4). Your analysis did not check for further markers affecting outcome (both prognostic and predictive markers) and other potential confounders. The findings you presents therefore need to be interpreted with caution. Further, such critical discussion (and discussion of other shortcomings of your study) is completely missing from the discussion.

Reviewer 2 Report

Authors investigated the contribution of NOX2 in the metabolism as well as it role in the switch of mesenchymal phenotype of GBM and GBM cell line U87MG. Comments and concerns about the study are listed as it follows

Major

  • Figure 1. (1) In the four panels of figure 1a, authors show the staining of NOX2 and GFAP protein by IHC. Since it is not stated in the text as well as it is hard to extrapolate this information from the pictures, can author provide with indication concerning the type of cells stained? Although, scale bars indicate 20um, cells in panel G4 appear larger than those in the other panels. Please discuss this. (2) Although, authors state in the figure caption that "White arrows indicate NOX2 and GFAP-positive cells", it is not clear whether this is colocalization. If I have understood correctly, it is not confocal microscopy. Maybe showing panels of single staining along with the merge image, could be oh help to better read the data. Otherwise, I would suggest turning to confocal microscopy. Finally, I would say, some of the arrows are wrongly placed.
  • Figure 2. Based on plots reported in Figure 2a, it seems that Oligomycin is not inhibiting ECAR (the curve keeps raising after Oligomycin addition), contrary to what 2DG does. This is opposite to what shown in the figure 2C, where ECAR decreases after administration of oligomycin. I would expect, at least in the control experiment, the same trend. Can author demonstrate that the Oligomycin is working. May be by measuring the level of ATP or ATP/ADP ratio? Indeed, since Oligomycin is an ATP synthase inhibitor, one would expect ATP to drop. Nevertheless, measurement of ECAR is hard since acidification of media is highly sensitive to pH changes, that can also depend on additional metabolites. Perhaps, additional chemical inhibitors can be used to determine pH changes. Can you discuss this point?
  • Figure 2. The right panels of figure 2a and 2C are confusing. First, for better reading of the data, it would worth it to label the four panels of figure 2 with specific letter (a to h) and clearly illustrate each panel in the figure caption. More important, can author explain why in NOX2 overexpressing cells both oligomycin and 2DG are not inhibiting acidification of extracellular media, e, ECAR is increasing instead of decreasing in the presence of the inhibitor? It seems that the effect of NOX2 overexpression overcomes that of chemical inhibition. Is that reasonable? Can you discuss it? Finally, always concerning this figure, according to authors' statement, immunoblots shown in figure 2 are representative images. So, why basal level of both NOX2 and HK2 are different in panel 2b and 2d? Perhaps it is worth it to show only the quantification with their statistics.
  • Curiously, always referring to figure 2, although the expression level of NOX2 after its overexpression is unexpectedly weak, it results in a huge increase of HK2. How can author explain this phenomenon?
  • Figure 3. Although, the GSK2795039 inhibitor reduces HK2 expression, what about the level of NOX2 under the same circumstances? Is it affected by GSK2795039? One more time, in figure 3a, the effect of oligomycin does not appear to be very effective in reducing ECAR level. I suppose, additional metabolites are influencing the read out of the experiments. Is it possible? Again, please label each panel of the figure.
  • Figure 4. It sounds weird to me that NOX staining shown in figure 1 and figure 4 are not fully comparable. In addition, although NOX staining of G4 samples in both figures is higher, in G2 and G3 panels of both figs NOX staining appears quite different. Why? Moreover, NH2 labeling is not fully convincing. Again, it is worth to turn to confocal and showing single panels along with the merged pictures. Finally, why cell size of panels G4 in both figures appear different if the scale bar is the same? Please, consider the use of the same fluorochrome/color for NOX across the manuscript. It is green in fig1 and red in fig 4.
  • Figure 5. Relatively to this figure, does the mRNA data correlate with protein expression level assayed by immunoblotting? Otherwise, do authors investigate the expression of Epithelial-mesenchymal-transition markers such as the members of the cadherin family or others? In addition, have authors investigate whether NOX2 overexpressing cells are more invasive or more resistant to chemotherapeutics than the control cells?

Minor

  • Line 92 to 95. It seems the verb in the sentence is missing.
  • Line 102. “…polyclonal mouse anti-NOX2”. I guess it is monoclonal. Please check your statement. Please, carefully check the information concerning all reagents used in the study for any inaccuracy.
  • Line 116 to 125. Please, rearrange and delete redundant parts of the paragraph.
  • Line 138. The term “against” doesn’t sound appropriate for overexpression. It is more appropriate for silencing or downregulation.
  • Section 2.5. Please note that, once stated the first time, there is no need to repeat catalog numbers, Company, etc. of the same reagent through the text. Please note that, in this section authors state that they used two different antibodies against NOX2 (polyclonal rabbit anti-NOX2 and a polyclonal mouse anti-NOX2) purchased from two different companies. However, it is not clear which one was really used in the study as well as displayed in figure 1. The same occurs for anti-GFAP. Please specify.
  • Line 254. Remove the extra “A”.
  • As authors focus on GFAP-positive samples, they should add explanation about the link between GFAP expression and tumor grade. It is important especially for readers not fully familiar with the experimental model. In addition, please indicate the full name of the protein, that is glial fibrillary acidic protein, in an appropriate section of the manuscript.
  • Line 414. Author state "Next, we measured the protein levels of COL5A1in 414 NOX2-positive cells..." I believe that the statement "to measure the protein level" when referring to immunofluorescence is not appropriate.

Reviewer 3 Report

The authors studied the role of NOX2 in the regulation of the glycolytic activity and the acquisition of a mesenchymal phenotype in human gliomas. 

The authors studied a small number of patients/cases and they grouped together diffuse gliomas and circumscribed glial/glioneuronal tumors (astrocytoma grade I (which corresponds to pilocytic astrocytoma) and ganglioglioma). Those two types of CNS tumors are totally different diseases, with very different prognoses (diffuse gliomas are consistently fatal whereas circumscribed glial/glioneuronal tumors can be cured by surgery alone).

Moreover, diffuse gliomas are classified according to the status (wild type or mutant) of IDH1/2 genes (2016 WHO classification). IDH gene mutations are associated with a significantly longer survival compared to IDH-wild type gliomas (such as glioblastomas). 

The conclusions are not supported by the experiments. Each subgroup comprises very few patients/cases. The number of cases studied should clearly appear in the material and methods section and in the result sections.

The methods used are very heterogeneous (PET scanning in patients, cell cultures, immunohistochemistry/immunofluorescence, survival data or gene expression data retrieved from the TCGA). 

It is not clear when univariate or multivariate analyses were performed.

Paragraph 2.5 is not clear. The techniques performed on FFPE material and on cultured cells should be presented separately.

Some sentences appear twice within the same paragraph (e.g., page 3, lines 116-125; page 9, paragraph 3.3).

WHO grade should appear as "grade" and not the letter G (beginning of introduction).

English needs editing.

There are typo errors.